# Splicing in a single neuron is coordinately controlled by RNA binding proteins and transcription factors

Morgan Thompson[1], Ryan Bixby[1], Robert Dalton[1], Alexa Vandenburg[1], John A Calarco[2], Adam D Norris[1]*

[1]Biological Sciences, Southern Methodist University, Dallas, United States; [2]Cell & Systems Biology, University of Toronto, Toronto, Canada

**Abstract** Single-cell transcriptomes are established by transcription factors (TFs), which determine a cell's gene-expression complement. Post-transcriptional regulation of single-cell transcriptomes, and the RNA binding proteins (RBPs) responsible, are more technically challenging to determine, and combinatorial TF-RBP coordination of single-cell transcriptomes remains unexplored. We used fluorescent reporters to visualize alternative splicing in single *Caenorhabditis elegans* neurons, identifying complex splicing patterns in the neuronal kinase *sad-1*. Most neurons express both isoforms, but the ALM mechanosensory neuron expresses only the exon-included isoform, while its developmental sister cell the BDU neuron expresses only the exon-skipped isoform. A cascade of three cell-specific TFs and two RBPs are combinatorially required for *sad-1* exon inclusion. Mechanistically, TFs combinatorially ensure expression of RBPs, which interact with *sad-1* pre-mRNA. Thus a combinatorial TF-RBP code controls single-neuron *sad-1* splicing. Additionally, we find 'phenotypic convergence,' previously observed for TFs, also applies to RBPs: different RBP combinations generate similar splicing outcomes in different neurons.
DOI: https://doi.org/10.7554/eLife.46726.001

*For correspondence:
adnorris@smu.edu

Competing interests: The authors declare that no competing interests exist.

## Introduction

The complement of genes expressed in an individual cell type controls its identity, development, and function. While transcriptional regulation is a major component of gene expression, post-transcriptional regulation can further shape cellular attributes by, for example, determining which gene isoforms are expressed in a cell. Much recent work has gone into cataloging gene expression networks in single cells, particularly those of specific neuronal types (*Tanay and Regev, 2017*; *Zeng and Sanes, 2017*). Molecular studies have also identified mechanisms by which transcription factors (TFs) shape gene expression networks in single neurons. Due to technical limitations, less is known about post-transcriptional regulation at the level of single neurons, or about the RNA binding proteins (RBPs) mediating post-transcriptional regulation (*Gracida et al., 2016*).

It is also unknown to what extent transcriptional and post-transcriptional gene regulatory networks are coordinated in single cells. A number of studies have identified individual RBPs that affect the splicing of a TF, thus altering the activity or specificity of that TF (*Calarco et al., 2009*; *Han et al., 2013*; *Linares et al., 2015*; *Raj et al., 2011*). These results suggest that there may be extensive cross-talk between transcriptional and post-transcriptional regulatory layers.

The nematode *Caenorhabditis elegans* has been used extensively as a model to reveal underlying principles by which TFs shape the transcriptomes of individual neurons. The worm's invariant cell lineage, coupled with genetic tools and a transparent body, enables systematic in vivo analysis of gene expression in single neurons, and identification of TFs responsible for cell-specific gene expression. This type of analysis has revealed a number of gene regulatory principles, including the concept

**eLife digest** All the cells in the human nervous system contain the same genetic information, and yet there are many kinds of neurons, each with different features and roles in the body. Proteins known as transcription factors help to establish this diversity by switching on different genes in different types of cells.

A mechanism known as RNA splicing, which is regulated by RNA binding proteins, can also provide another layer of regulation. When a gene is switched on, a faithful copy of its sequence is produced in the form of an RNA molecule, which will then be 'read' to create a protein. However, the RNA molecules may first be processed to create templates that can differ between cell types: this means that a single gene can code for slightly different proteins, some of them specific to a given cell type. Yet, very little is known about how RNA splicing can generate more diversity in the nervous system.

To investigate, Thompson et al. developed a fluorescent reporter system that helped them track how the RNA of a gene called *sad-1* is spliced in individual neurons of the worm *Caenorhabditis elegans*. This showed that *sad-1* was turned on in all neurons, but the particular spliced versions varied widely between different types of nerve cells.

Additional experiments combined old school and cutting-edge genetics technics such as CRISPR/Cas9 to identify the proteins that control the splicing of *sad-1* in different kinds of neurons. Despite not directly participating in RNA splicing, a number of transcription factors were shown to be involved. These molecular switches were turning on genes that code for RNA binding proteins differently between types of neurons, which in turn led *sad-1* to be spliced according to neuron-specific patterns.

The findings by Thompson et al. could provide some insight into how mammals can establish many types of neurons; however, a technical hurdle stands in the way of this line of research, as it is still difficult to detect splicing in single neurons in these species.

DOI: https://doi.org/10.7554/eLife.46726.002

of a 'combinatorial code' of TFs which can be re-used in different neuron types, with particular combinations of TFs determining specific cell fates (*Gendrel et al., 2016*; *Gordon and Hobert, 2015*; *Pereira et al., 2015*). Another example is the concept of 'phenotypic convergence' by which various neurons express similar gene networks but the TFs driving the networks are different for each neuron type (*Gendrel et al., 2016*; *Pereira et al., 2015*). These principles appear to apply to the nervous systems of other organisms as well (*Konstantinides et al., 2018*). However, it remains unknown whether similar mechanistic principles apply to post-transcriptional regulation by RBPs in the nervous system.

Here we use single-cell in vivo fluorescent splicing reporters to investigate the cell-specific splicing of *sad-1*, a conserved neuronal kinase. The *C. elegans sad-1* gene encodes two isoforms that differ in their ability to interact with the F-actin binding protein NAB-1/Neurabin (*Hung et al., 2007*), and have different roles in synapse formation and development (*Kim et al., 2010*). We find that *sad-1* undergoes unique splicing patterns in various neuron types, and that developmentally-related cell types (the ALM touch-sensing neuron and the BDU neuron) exhibit opposing patterns of splicing (exon inclusion vs. exon skipping). A combination of unbiased genetic screens and candidate targeted mutations identified a cascade of three cell-fate determining TFs and two neuronal RBPs required for proper splicing of *sad-1* in ALM neurons. Mechanistic dissection revealed that the three TFs function to establish cell-specific expression of the two RBPs in the ALM neuron, and that the two RBPs in turn directly bind to *sad-1* intronic regions to mediate exon inclusion in the ALM neuron. Finally, we find that in other neuron types, similar principles apply but with different combinations of TFs and RBPs mediating *sad-1* exon inclusion. These results indicate that neuronal RBPs, like TFs, are employed in a combinatorial code to shape neuron-specific splicing patterns, and demonstrate phenotypic convergence by which different RBPs mediate similar splicing outcomes in various neurons.

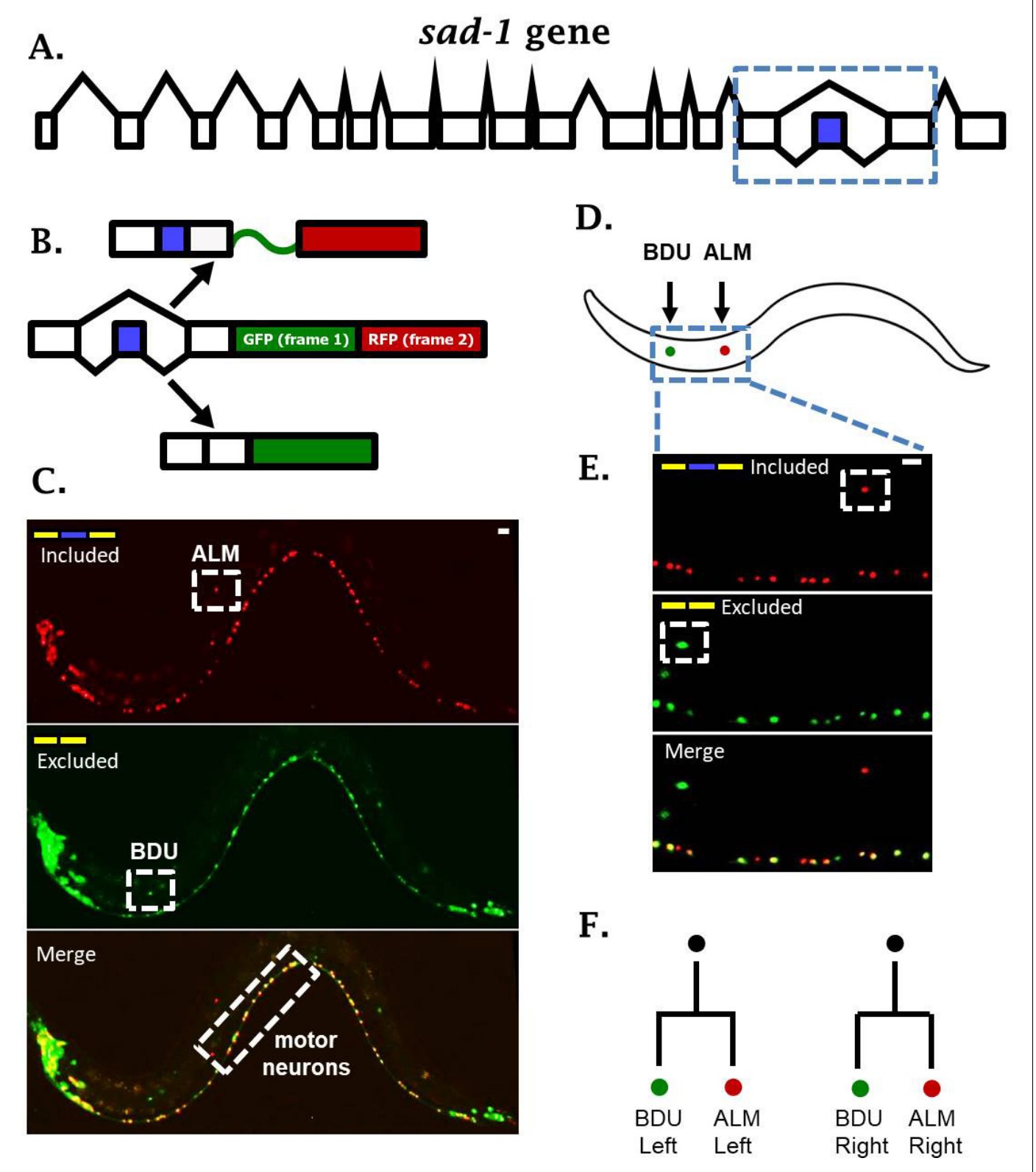

**Figure 1.** *sad-1* is alternatively spliced in single neurons. (A) The *sad-1* gene. Alternative cassette exon in blue. (B) Two-color splicing reporter schematic for *sad-1* cassette exon. The cassette exon encodes a + 1 nt frameshift so that when skipped, GFP is produced with an in frame stop codon. When skipped, GFP is read out of frame without stop codons, followed by in-frame translation of RFP. (C) Whole worm fluorescent micrograph demonstrating both exon inclusion (RFP) and skipping (GFP) in many neurons, while certain neurons express only the included (ALM) of skipped (BDU) isoforms. (D–E)
*Figure 1 continued on next page*

*Figure 1 continued*

Higher magnification focusing on ALM and BDU neurons. (**F**) BDU and ALM are both paired neurons present on the left and right side of the worm. Each BDU neuron is a sister cell to an ALM neuron, derived from the same neuroblast. Scale bar represents 10 µm.

DOI: https://doi.org/10.7554/eLife.46726.003

The following figure supplement is available for figure 1:

**Figure supplement 1.** Gross quantification of fraction of neurons expressing included (red), skipped (green), or both (yellow) isoforms of *sad-1*, in either ventral nerve cord, head, or tail.

DOI: https://doi.org/10.7554/eLife.46726.004

## Results

### Alternative splicing of the neuronal kinase *sad-1* in specific cell types

To identify alternative splicing regulation in individual neuronal cell types, we created two-color splicing reporters that provide a fluorescent readout of splicing regulation in vivo in single cells (*Kuroyanagi et al., 2006*; *Orengo et al., 2006*). A minigene representing an alternative splicing event of interest is cloned upstream of a dual GFP/RFP cassette (*Figure 1A–B*). The GFP and RFP coding sequences reside in alternative reading frames. The alternative exon is engineered to shift the reading frame by +1 nucleotide such that splicing of the alternative exon determines the reading frame, and therefore the translation of GFP versus RFP. Application of two-color fluorescent reporters to transparent organisms such as *C. elegans* enables in vivo imaging of alternative splicing in individual cells. We have created reporters for splicing events in a number of neuronal genes, and uncovered a rich variety of splicing patterns in single neurons (*Norris et al., 2014*).

One intriguing example of neuron-specific alternative splicing is in the conserved neuronal kinase *sad-1*, which plays important roles in neuronal development in both worms and mice (*Kim et al., 2008*; *Kishi et al., 2005*). In *C. elegans, sad-1* is encoded by seventeen exons, and the fifteenth exon is an alternative cassette-type exon (*Figure 1A*). Alternative splicing of this exon changes the coding sequence and length of the *sad-1* C-terminus (*Kim et al., 2010*). This presents an interesting parallel with mice and human genomes, which encode two separate genes homologous to *sad-1* (SAD-A and SAD-B) that are nearly identical except for their C-terminal coding sequence and length.

A two-color splicing reporter for *sad-1* in *C. elegans* revealed that many neurons express both the skipped and included isoforms (*Figure 1C*, *Figure 1—figure supplement 1*). For example, motor neurons in the ventral nerve cord express both isoforms of *sad-1* (*Figure 1C*). On the other hand, the ALM touch-sensing neuron expresses only the included isoform, while the BDU neuron, which is the sister cell to the ALM neuron, expresses only the skipped isoform (*Figure 1C–F*). While different neurons exhibit differences in *sad-1* splicing, the splicing pattern in a given neuron is reproducible and invariant from one animal to the next, suggesting that *sad-1* splicing in various neurons is under strict regulatory control. These results led us to ask how ALM and BDU neurons, which are developmentally related (*Figure 1F*) and share a number of anatomical and gene-expression features, specify opposite splicing regimes.

### Forward genetic screen identifies a trio of fate-determining TFs affecting *sad-1* splicing in the ALM neuron

To identify regulators of *sad-1* splicing in the ALM touch neuron, we performed an unbiased forward genetic screen. Parental worms harboring the *sad-1* splicing reporter were mutagenized with EMS. We then screened for $F_2$ animals (potential homozygotes) with aberrant expression of the skipped (GFP) isoform in the ALM neuron (*Figure 2A*). This screen identified three distinct loci that transform the splicing pattern from the ALM neuron pattern (full exon inclusion) to resemble the pattern in their BDU sister cells (full exon skipping).

Whole-genome resequencing of the mutant strains identified loss-of-function mutations in three conserved TFs: *unc-86*, *mec-3*, and *alr-1* (*Figure 2B–F*, *Figure 2—figure supplement 1*). All three genes have previously been identified as key regulators of touch-neuron cell fate (*Gordon and Hobert, 2015*; *Topalidou et al., 2011*). The three TFs function in a transcriptional cascade ensuring cell-specific expression of *mec-3* in touch neurons, which then results in expression of a battery of touch-neuron specific genes (*Figure 2G*). Loss of the TF *mec-3* results in touch neurons (ALMs)

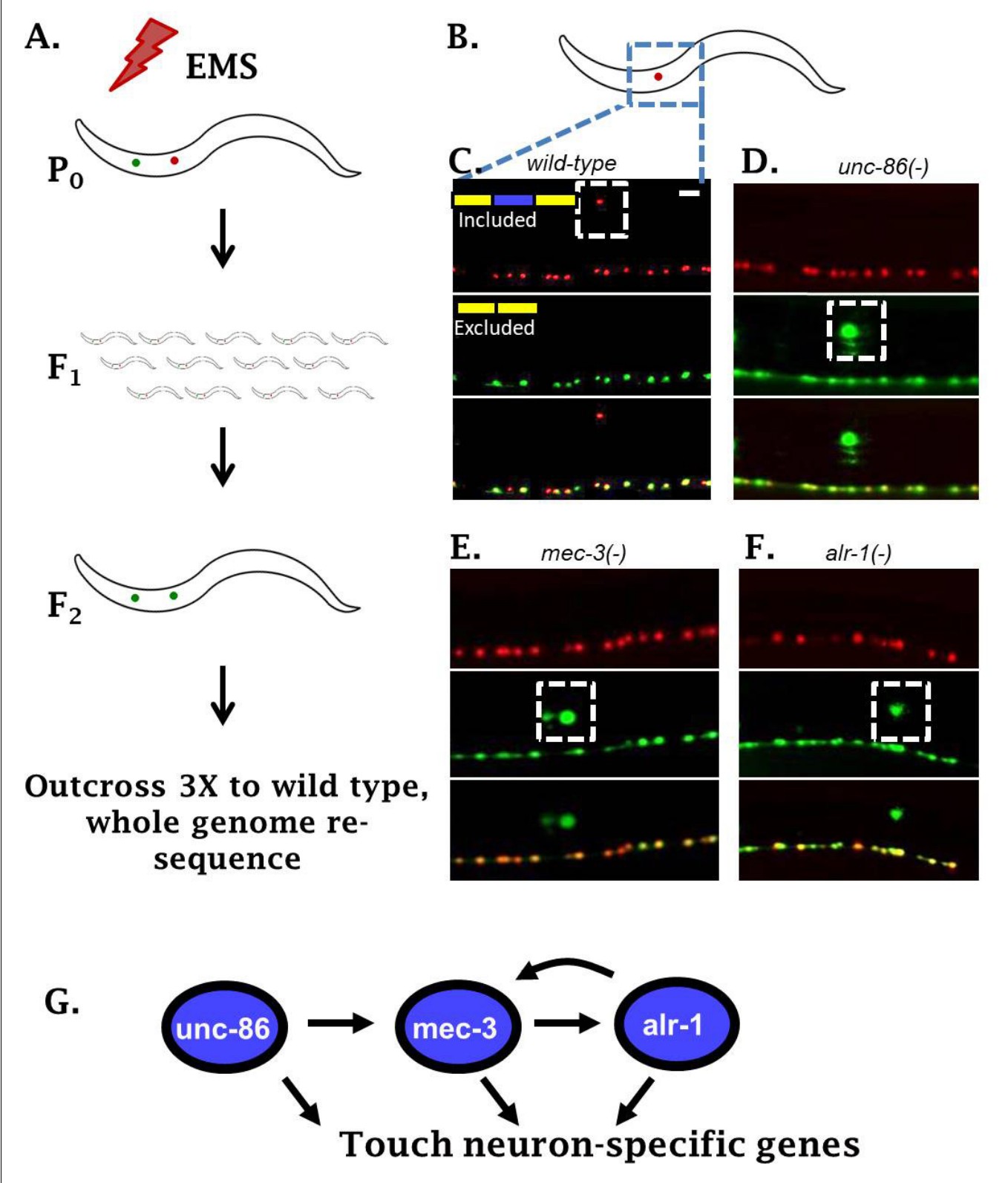

**Figure 2.** Genetic screen identifies neuronal TFs affecting *sad-1* splicing in the ALM neuron. (A) Schematic of forward genetic screen to identify regulators of *sad-1* splicing in the ALM touch neuron. (B–F) ALM neurons (dashed boxes) shift from complete inclusion (RFP) to skipping (GFP) in *unc-86 (e1416)*, *mec-3(e1338)*, or *alr-1(oy42)* TF mutants. Splicing phenotypes fully penetrant (n = 50 animals) (G) Previously-identified roles of the three TFs in a transcriptional cascade to control touch neuron gene expression. Scale bar represents 10 µm.

*Figure 2 continued on next page*

eLIFE Research article

Developmental Biology | Genetics and Genomics

*Figure 2 continued*

DOI: https://doi.org/10.7554/eLife.46726.005

The following figure supplement is available for figure 2:

**Figure supplement 1.** TF alleles identified in genetic screen cause sad-1 splicing defects in ALM: unc-86(csb9), mec-3(csb10), alr-1(csb11).

DOI: https://doi.org/10.7554/eLife.46726.006

adopting certain gene-expression characteristics of their sister cells (BDUs) (*Gordon and Hobert, 2015*), mirroring our observation that loss of *mec-3* transforms *sad-1* splicing from an ALM (exon 15 included) to a BDU (exon 15 skipped) pattern.

Previous work demonstrates that the MEC-3 TF is expressed only in touch neurons, while UNC-86 and ALR-1 are expressed in various neuron types (*Topalidou et al., 2011*). However, we find that *unc-86* and *alr-1* mutants affect *sad-1* splicing only in the touch neurons (*Figure 2D–F*). This is in accordance with previous work indicating that a major function of *unc-86* and *alr-1* in touch neurons is to combinatorially ensure appropriate expression of *mec-3*, and that all three TFs are needed for proper differentiation of touch neurons (*Topalidou et al., 2011*). We therefore conclude that the combinatorial activity of all three TFs is required for proper *sad-1* splicing in the ALM neuron.

## A pair of RNA binding proteins regulates *sad-1* splicing in the ALM neuron

We were surprised to identify TFs, but not RBPs, in our forward genetic screen for regulators of *sad-1* alternative splicing. We hypothesized that multiple RBPs might co-regulate *sad-1* alternative splicing in the ALM neuron and therefore mutations in individual RBPs might result in mild splicing defects. We therefore examined the sequence surrounding the *sad-1* alternative exon for conserved *cis*-elements corresponding to known in vitro RBP sequence preferences (*Ray et al., 2013*). We identified three candidate elements: one corresponding to the *mbl-1*/Mbnl1 consensus binding motif, and two corresponding to the *mec-8/RBMS* motif (*Figure 3A–C*).

To test whether these RBPs affect *sad-1* alternative splicing, we created deletions for each gene with CRISPR/Cas9 (*Norris et al., 2017*). Both *mec-8* and *mbl-1* mutants result in aberrant *sad-1* splicing in the ALM neuron, displaying partial skipping and partial inclusion (*Figure 3D–F*, *Figure 3—figure supplement 1*). As in the case of the TF mutants, *mec-8* mutants affect *sad-1* splicing specifically in the ALM neurons, whereas *mbl-1* mutants affect *sad-1* splicing in ALM neurons as well as specific neurons in the ventral nerve cord (see Figure 6, below). To verify that the phenotypes of our CRISPR mutants were on-target effects, we crossed the *sad-1* splicing reporter into existing alleles for *mec-8* (*e398*, premature stop codon [*Davies et al., 1999*; *Lundquist et al., 1996*]) and *mbl-1* (*wy560,* large deletion affecting multiple genes including *mbl-1* [*Spilker et al., 2012*]). We found these alleles to affect splicing of *sad-1* exactly as our CRISPR mutations (*Figure 3—figure supplements 1–2*).

Whereas TF mutants result in full skipping of the *sad-1* alternative exon, RBP mutants result in only partial skipping. This provides a probable explanation for not identifying these RBPs in our genetic screen: partial exon skipping leads to dim GFP expression, which is not sufficiently bright to be noticed upon brief visual inspection. We therefore tested whether simultaneous loss of both RBPs recapitulates the full skipping of *sad-1* exon 15 observed in TF mutants. We created *mec-8; mbl-1* double mutants expressing the *sad-1* splicing reporter. These double mutants result in complete loss of *sad-1* exon inclusion in the ALM neuron, recapitulating the splicing phenotype of the single TF mutants (*Figure 3G*). These results led us to hypothesize that the TFs identified in our screen exert their effects on *sad-1* splicing by controlling expression of both *mec-8* and *mbl-1*.

## TFs affecting *sad-1* splicing are required for RBP expression in the ALM neuron

To examine whether the neuronal TFs alter expression of *mec-8* and *mbl-1* RBPs in the ALM neuron, we created reporter lines for each RBP. To this end, each RBP was C-terminally tagged in a fosmid containing large regions of surrounding genomic context (*Poser et al., 2008*; *Spilker et al., 2012*) (*Figure 4A–E*). Compared to traditional transgenic reporters, fosmids are more likely to contain all regulatory information needed to drive normal expression of the gene in question. This is demonstrated in the case of the *mec-8* RBP. The classical *mec-8::GFP* promoter fusion drives expression in

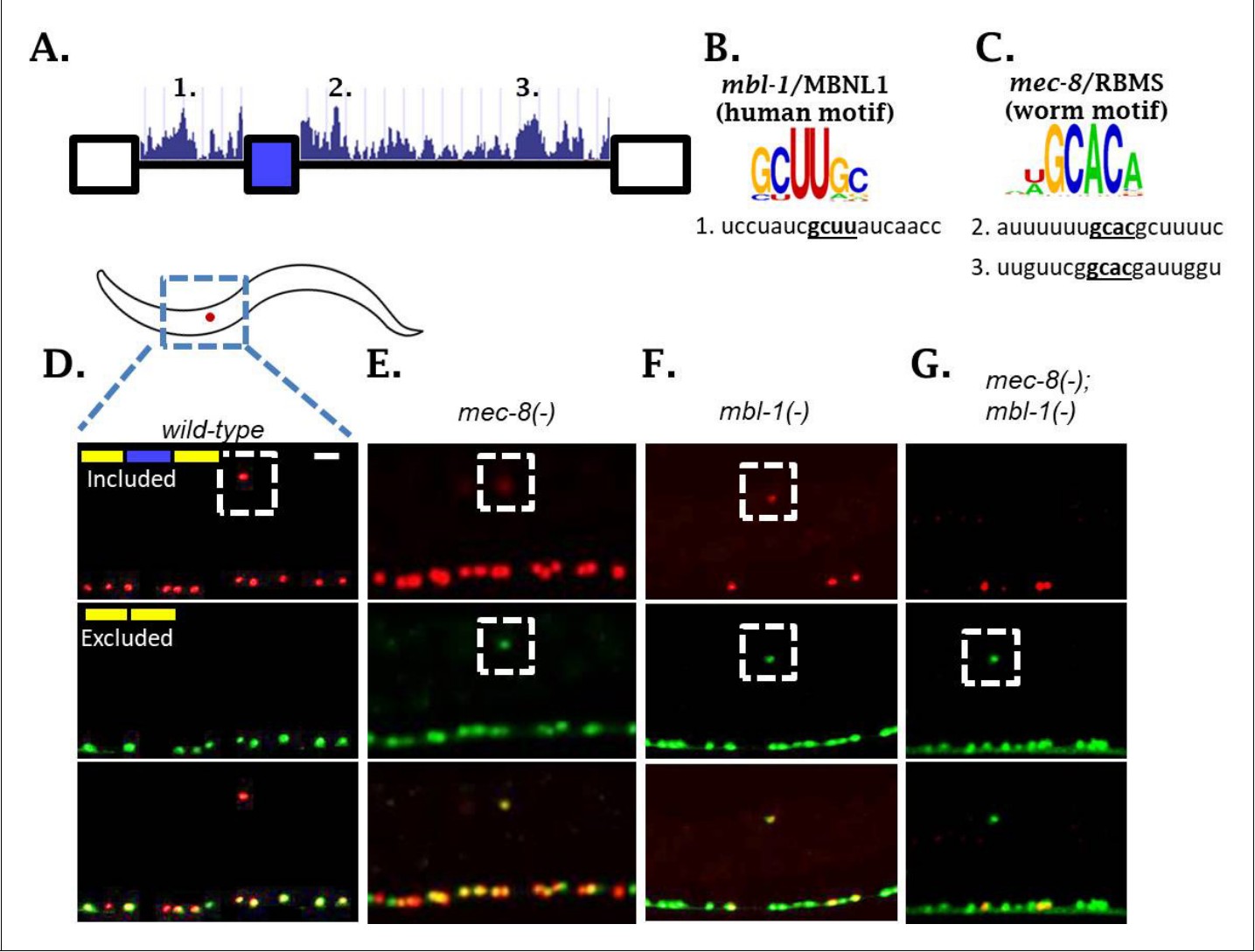

**Figure 3.** Two neuronal RBPs combinatorially control *sad-1* splicing in ALM neurons. (**A**) Conservation scores in the introns surrounding *sad-1* exon 15, basewise phyloP26way comparison of 26 nematode species (***Hubisz et al., 2011***). Numbers 1–3 indicate consensus binding motifs for *mbl-1* and *mec-8* displayed in B-C. (**B–C**) *cis*-elements matching consensus binding motifs for *mbl-1* and *mec-8*. (**D–F**) *mec-8* and *mbl-1* mutants both cause a partial loss of *sad-1* exon inclusion. (**G**) *mec-8; mbl-1* double mutants cause complete loss of exon inclusion, phenocopying the TF mutants. Splicing phenotypes fully penetrant (n = 50 animals) Scale bar represents 10 μm.

DOI: https://doi.org/10.7554/eLife.46726.007

The following figure supplements are available for figure 3:

**Figure supplement 1.** Deletion alleles used in this study, in addition to canonical mutations and mutations identified in our forward genetic screen.
DOI: https://doi.org/10.7554/eLife.46726.008

**Figure supplement 2.** Canonical RBP alleles of mec-8 and mbl-1 affect sad-1 splicing similarly to CRISPR deletions of mec-8 and mbl-1.
DOI: https://doi.org/10.7554/eLife.46726.009

a number of cells, but not in the ALM neuron (***Figure 4—figure supplement 1***) (***Spike et al., 2002***). On the other hand, we detected expression of the *mec-8* fosmid reporter in many of the same cells, both neuronal and non-neuronal, plus strong expression in the ALM neuron (***Figure 4A–B***). A similar fosmid reporter for *mbl-1* likewise exhibits expression in the ALM neuron, as well as many other neurons in the nervous system (***Figure 4D***, ***Figure 4—figure supplement 1***). This is in line with previous reports on *mbl-1* expression (***Spilker et al., 2012***).

We tested expression of our reporters in the context of a *mec-3* mutant to determine whether expression of *mec-8* and *mbl-1* in ALM neurons depends on the TF cascade uncovered in our screen.

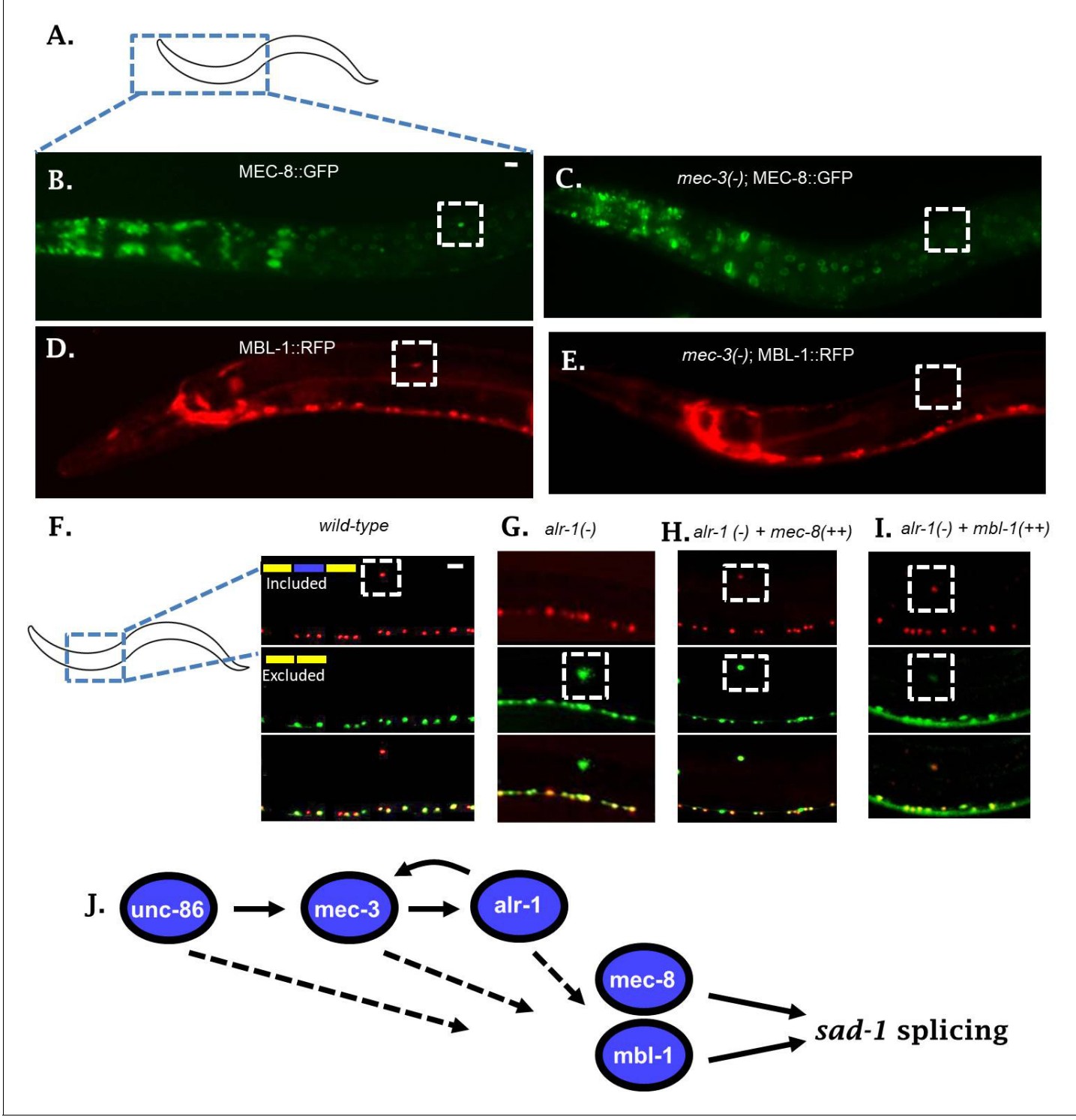

**Figure 4.** Neuronal TFs establish expression of both *mec-8* and *mbl-1* to mediate splicing of *sad-1* in ALM neurons. (**A–B**) A *mec-8* translational GFP fosmid reporter reveals strong expression in ALM neuron (strong expression in 28/31 = 90% of animals inspected). (**C**) In a *mec-3* TF mutant, *mec-8* expression is absent specifically in ALM (no detectable expression in 43/50 = 86%, dim expression in 7/50 = 14% of animals inspected). (**D**) *mbl-1* translational RFP fosmid reporter is expressed in ALM neuron (strong expression in 19/20 = 95% of animals inspected). (**E**) In a *mec-3* mutant, *mbl-1* expression is absent specifically in ALM (no detectable expression in 19/21 = 90%, dim expression in 2/21 = 10% of animals inspected). (**F–G**) Aberrant splicing of *sad-1* in *alr-1* TF mutants is partially rescued by over-expression of either *mec-8* (6/6 animals examined) or *mbl-1* (6/7 animals examined) RBPs (**H–I**). Scale bar represents 10 μm.

DOI: https://doi.org/10.7554/eLife.46726.010

*Figure 4 continued on next page*

eLIFE Research article

Developmental Biology | Genetics and Genomics

**Figure supplement 1.** Transcriptional and translational reporters for MEC-8 and MBL-1.
DOI: https://doi.org/10.7554/eLife.46726.011
**Figure supplement 2.** ALR-1 ChIP peaks in *mec-8* and *mbl-1* promoters.
DOI: https://doi.org/10.7554/eLife.46726.012
**Figure supplement 3.** Double heterozygotes for TFs and RBPs display moderate defects in *sad-1* splicing in ALM neurons.
DOI: https://doi.org/10.7554/eLife.46726.013

The *mec-3* TF is expressed only in touch neurons, and therefore we would expect *mec-3* mutants to affect RBP expression only in the touch neurons. Indeed, in *mec-3* mutants, expression of both *mec-8* and *mbl-1* RBPs are abolished in the ALM neuron, while expression in the surrounding neurons and tissues remains unchanged (*Figure 4B–E*). Together these results indicate that the expression of *mec-8* and *mbl-1* RBPs are under the control of neuron subtype-specific TFs.

To examine whether *mec-8* and *mbl-1* RBPs might be under direct transcriptional control by one or more of the TFs, we used existing ChIP data for ALR-1 (*Niu et al., 2011*), in vitro derived consensus binding motifs for UNC-86 (*Weirauch et al., 2014*), and a previously-defined UNC-86/MEC-3 heterodimer binding motif (*Röhrig et al., 2000*; *Xue et al., 1993*). We did not find conserved UNC-86 binding motifs or an UNC-86/MEC-3 heterodimer binding motif in the promoters for *mec-8* or *mbl-1*, but did find ALR-1 ChIP peaks in both promoters (*Figure 4—figure supplement 2*). This data suggests that *alr-1* may directly control transcription of *mec-8* and *mbl-1* RBPs.

## TFs affect *sad-1* splicing by controlling RBP expression in the ALM neuron

The observations that (1) *mec-8; mbl-1* RBP double mutants recapitulate the phenotype of the TF mutants, and (2) the TFs are necessary for expression of both RBPs in the ALM neuron, together suggest that the splicing defects in the TF mutants are mediated by effects on expression of the two RBPs. Further support for this hypothesis arose indirectly in the course of crossing TF and RBP mutants together. We found that while TF or RBP mutant heterozygotes exhibit normal *sad-1* splicing in the ALM neuron, double heterozygotes (for example *alr-1/+; mbl-1/+*, or *mec-3/+; mec-8/+*) exhibit partial exon skipping in ALM, similar to the RBP single mutants (*Figure 4—figure supplement 3*). Such 'non-allelic non-complementation' is often interpreted to mean that the two genes function in the same complex, or, more likely in this case, function in the same pathway (*Yook et al., 2001*). This indirect evidence further suggests that the TFs and RBPs affect *sad-1* splicing as part of the same molecular pathway.

If *sad-1* splicing is controlled in a linear pathway as suggested by the above series of experiments, with upstream TFs affecting RBP expression in the ALM neuron, then over-expressing an RBP in the context of a TF mutant should partially restore splicing in ALM. To test this hypothesis we created a strain over-expressing a *mec-8* transgene specifically in the touch neurons (*pmec-3::mec-8*). When introduced into an *alr-1* mutant, this transgene partially rescues the splicing of *sad-1* in the ALM neuron (*Figure 4F–H*). Likewise, over-expression of *mbl-1* in an *alr-1* mutant partially rescues splicing in the ALM neuron (*Figure 4I*). These results further support a linear gene regulatory pathway in which neuronal fate-determining TFs control neuron-specific expression of RBPs, which then control alternative splicing of *sad-1* (*Figure 4J*).

## RBPs directly mediate *sad-1* exon inclusion through interactions with surrounding introns

To test whether *mec-8* and *mbl-1* directly affect splicing by binding to the *sad-1* pre-mRNA, we created two-color splicing reporters in which the putative *mec-8* or *mbl-1* cis-elements are mutated (*Figure 3A* and *Figure 5*). If the RBPs act directly by binding the *cis*-element, then mutation of the *cis*-element should affect the splicing pattern in a manner resembling the wild-type splicing reporter in the context of the RBP deletion mutant. If the RBPs act indirectly, mutating the *cis*-element should have no effect on the splicing pattern.

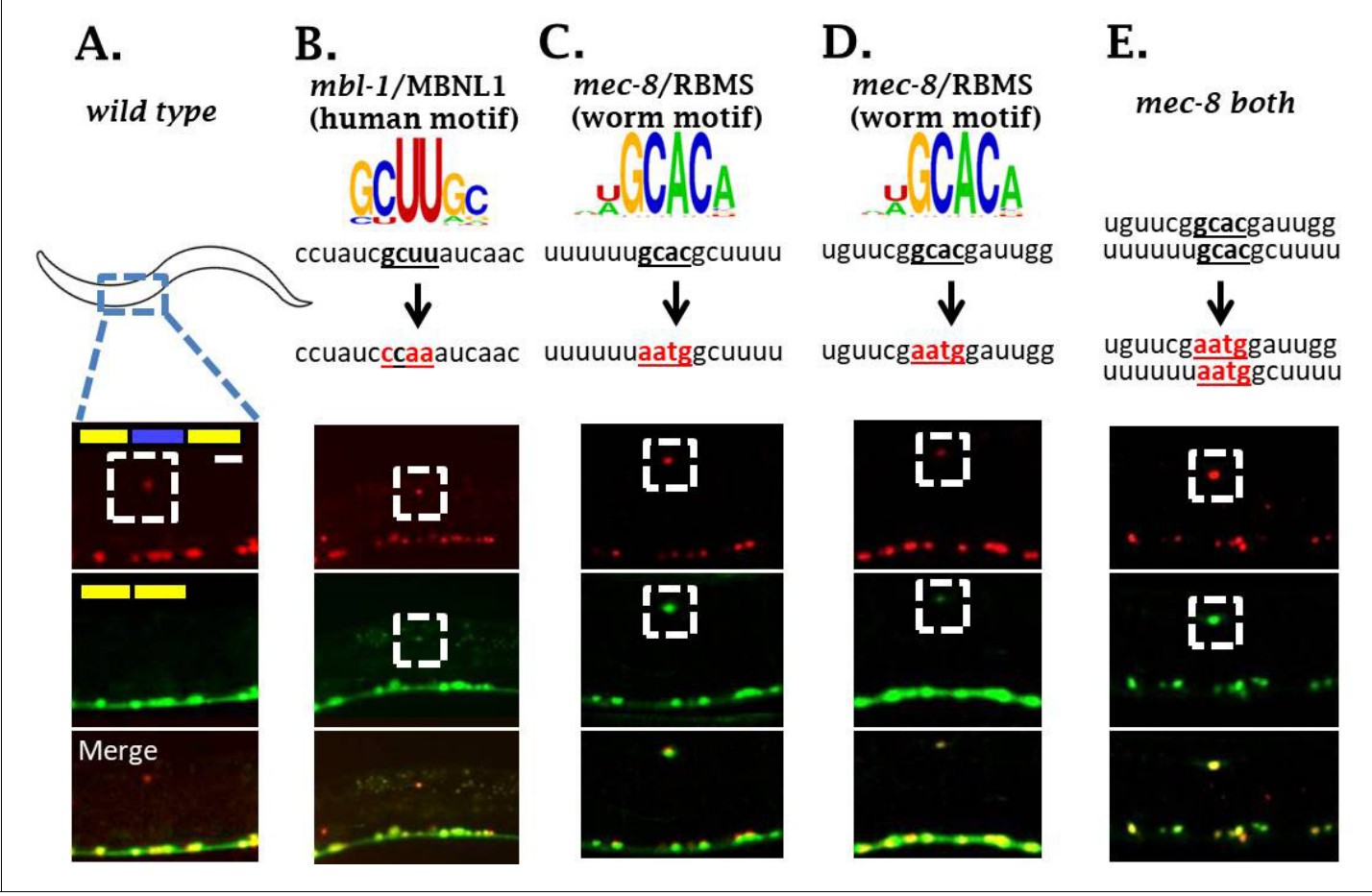

**Figure 5.** *mbl-1* and *mec-8* affect *sad-1* splicing by direct interaction with *sad-1* introns. (A–B) Mutation of *mbl-1* consensus sequence in *sad-1* splicing reporter results in aberrant splicing in ALM neurons that phenocopies an *mbl-1* mutant. (C–E) Mutation of either *mec-8* binding motif, or both simultaneously, likewise results in aberrant *sad-1* splicing in ALM neurons. ALM splicing phenotypes fully penetrant (n = 25 animals) Scale bar represents 10 µm.

DOI: https://doi.org/10.7554/eLife.46726.014

The following figure supplement is available for figure 5:

**Figure supplement 1.** RBP overexpression does not rescue *sad-1* splicing defects of cognate *cis*-element mutant reporters (failure to rescue in n = 20 animals for each condition).

DOI: https://doi.org/10.7554/eLife.46726.015

Mutation of the *mbl-1 cis*-element resulted in ALM neurons with altered *sad-1* splicing in which the exon is partially skipped and partially included (*Figure 5A–B*). This recapitulates the phenotype of *mbl-1* null mutations (*Figure 3F*), suggesting that *mbl-1* exerts its effects on splicing directly through binding a conserved *cis*-element in the upstream intron.

We identified two consensus *mec-8* binding motifs in conserved regions in the intron downstream of the cassette exon. We therefore created splicing reporters mutant for both *cis*-elements as well as for each element individually. The splicing reporter mutant for both elements recapitulates the splicing phenotype of *mec-8* null mutants (*Figure 5E*). Likewise, mutating either *mec-8* binding site in isolation recapitulates a *mec-8* null mutation (*Figure 3E* and *Figure 5C–D*), suggesting that *mec-8* binding to both *cis*-elements is required for appropriate *sad-1* splicing.

We tested whether mutation of a putative *cis*-element could be rescued by over-expression of its cognate RBP, and found that *cis*-element mutants were not rescued by RBP over-expression (*Figure 5—figure supplement 1*), providing further evidence that the RBPs act directly on the *sad-1* pre-mRNA. Together these results indicate that *mec-8* and *mbl-1* RBPs combinatorially ensure *sad-1* exon inclusion in ALM neurons through direct interactions with the neighboring introns.

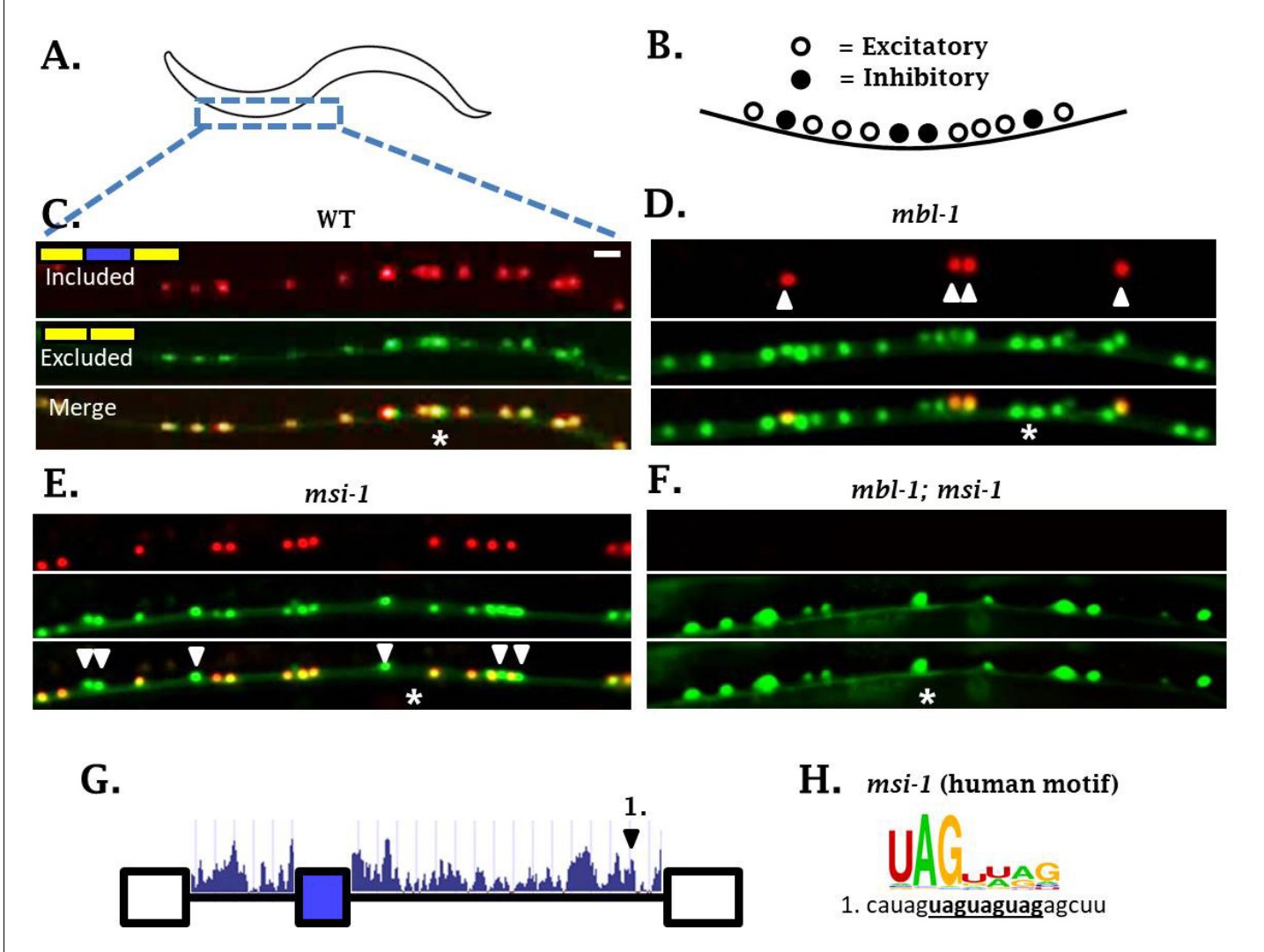

**Figure 6.** *sad-1* splicing in motor neurons of the ventral nerve cord is controlled by *mbl-1* and *msi-1* RBPs. (A–C) In wild-type worms, *sad-1* is partially included in both excitatory and inhibitory motor neurons. (D) In *mbl-1* mutants, exon inclusion is lost in excitatory motor neurons, but remains in inhibitory motor neurons (arrowheads). (E) *msi-1* mutants lose exon inclusion in inhibitory motor neurons (arrowheads) but not in excitatory motor neurons. (F) *mbl-1; msi-1* double mutants lose exon inclusion in all motor neurons in the ventral nerve cord. Splicing phenotypes in ventral nerve cord invariant (n = 15 animals) (G) Conservation scores (determined as in *Figure 3A*) in the introns surrounding *sad-1* exon 15. Number one indicates consensus binding motifs for *msi-1*. (H) *cis*-elements matching consensus binding motifs for *msi-1*. Asterisk indicates anterior-posterior position of ALM neuron as anatomical reference. Splicing phenotypes fully penetrant (n = 50 animals). Scale bar represents 10 μm.

DOI: https://doi.org/10.7554/eLife.46726.016

The following figure supplements are available for figure 6:

**Figure supplement 1.** MBL-1, visualized by a translational RFP fusion is expressed specifically in the excitatory cholinergic neurons of the ventral nerve cord, visualized by an *unc-17::BFP* promoter reporter.

DOI: https://doi.org/10.7554/eLife.46726.017

**Figure supplement 2.** *sad-1* splicing is controlled by distinct RBPs and TFs in ventral nerve cord motor neurons.

DOI: https://doi.org/10.7554/eLife.46726.018

## *sad-1* splicing in other neuron types is controlled through both distinct and overlapping mechanisms

Having identified regulatory mechanisms controlling *sad-1* splicing in the ALM neuron, we next wondered whether similar principles apply in other neuron types. Most neurons besides the ALM and BDU neurons express both skipped and included *sad-1* isoforms. This could represent the neuronal

'ground state' of splicing in the absence of cell-specific splicing regulators. On the other hand, our observations that loss of both *mec-8* and *mbl-1* in the ALM neuron results in full exon skipping suggest that the ground state may be complete exon skipping. This hypothesis predicts that other neurons in which *sad-1* is partially included express one or more RBPs mediating exon inclusion.

In the course of examining *sad-1* splicing in ALM neurons, we noticed that *mbl-1* mutants affect *sad-1* splicing not only in ALM, but also in the excitatory cholinergic motor neurons of the ventral nerve cord (*Figure 6A–D*). Whereas *mbl-1* mutants cause a change in *sad-1* splicing from full inclusion to partial inclusion in ALM neurons, in excitatory motor neurons *mbl-1* mutants shift from partial inclusion to no inclusion (*Figure 6C–D*). On the other hand, the inhibitory motor neurons remain unaffected in *mbl-1* mutants, expressing both the included and skipped isoforms (*Figure 6D*, arrowheads). This is consistent with our *mbl-1* gene expression reporter, which reveals expression of *mbl-1* in the excitatory motor neurons, but not in the inhibitory motor neurons (*Figure 6—figure supplement 1*).

We did not detect *mec-8* expression in motor neurons of the ventral nerve cord, and *mec-8* mutants had no effect on splicing of *sad-1* in motor neurons (*Figures 3E* and *4B*). It therefore seems that in neurons expressing *mbl-1* such as excitatory motor neurons, the presence of *mbl-1* mediates partial exon inclusion. In neurons expressing both *mbl-1* and *mec-8* such as ALM touch neurons, the two RBPs together mediate full inclusion.

In *mbl-1* mutants, *sad-1* exon inclusion is lost in excitatory neurons but remains in inhibitory motor neurons. We therefore wondered whether there was an additional RBP expressed in inhibitory motor neurons mediating *sad-1* inclusion. *mec-8* was ruled out because it is not expressed in inhibitory motor neurons and does not affect *sad-1* splicing in the nerve cord. On the other hand, the RBP *msi-1*/Musashi has been reported to be expressed in inhibitory but not excitatory neurons of the nerve cord (*Yoda et al., 2000*), which is a mutually exclusive pattern with *mbl-1*. We therefore tested *msi-1*

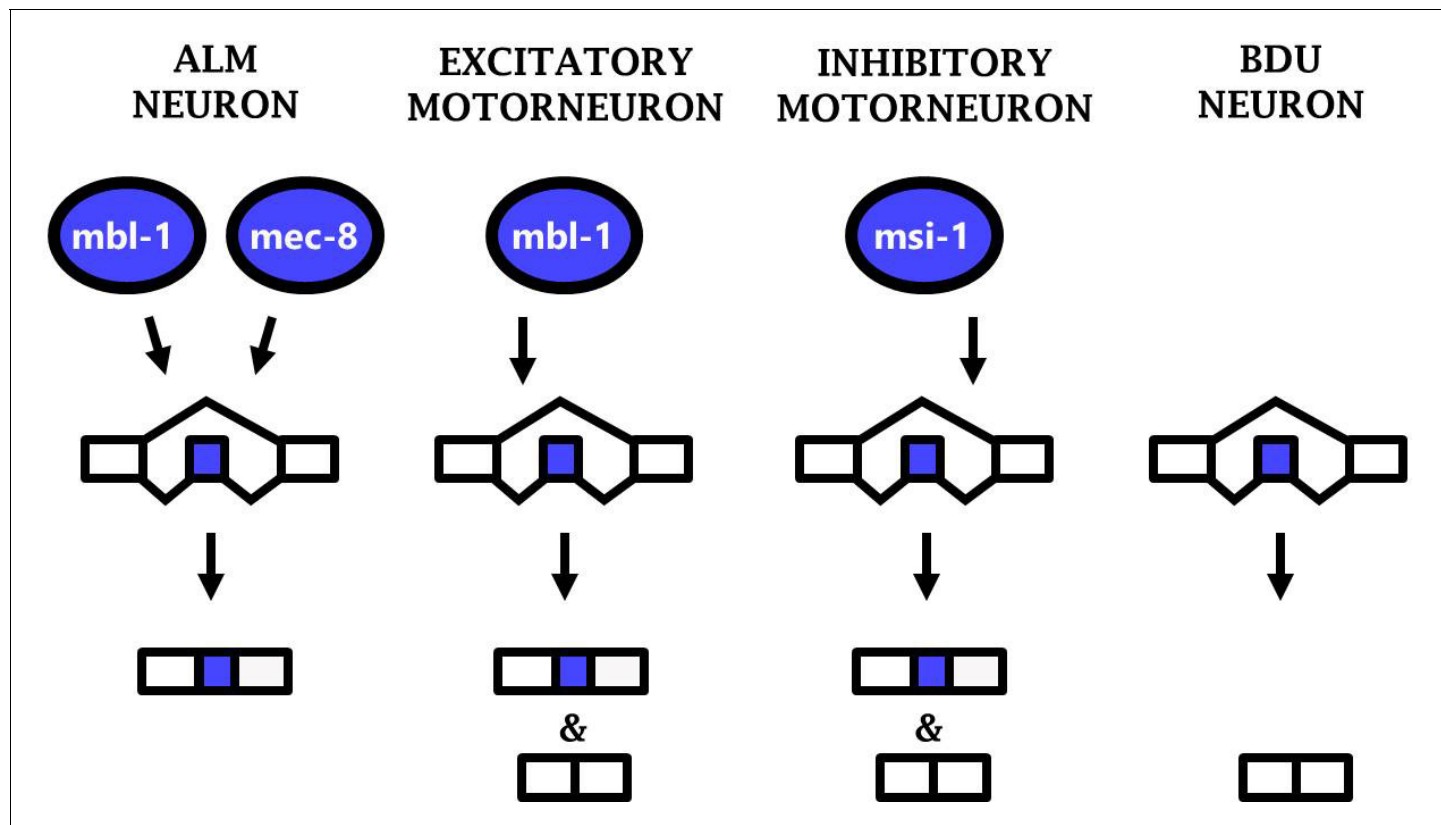

**Figure 7.** Phenotypic convergence at the level of splicing regulation. Different RBPs act in different neuron types to carry out the common function of mediating *sad-1* exon inclusion.
DOI: https://doi.org/10.7554/eLife.46726.019

as a candidate for the RBP mediating *sad-1* exon inclusion in the inhibitory motor neurons. We generated a *msi-1* deletion mutant, which shows loss of *sad-1* inclusion specifically in the inhibitory motor neurons (*Figure 6E*). Furthermore, *msi-1; mbl-1* double mutants result in complete loss of exon inclusion in the ventral nerve cord (*Figure 6F*). These results indicate that *mbl-1* and *msi-1* act in distinct cell types to achieve partial *sad-1* exon inclusion throughout the ventral nerve cord.

We suspect that *msi-1*, like *mbl-1* and *mec-8*, directly affects *sad-1* splicing by binding in the intronic regions surrounding the alternative exon. in vitro experiments have identified a UAG motif (*Figure 6H*) (*Ray et al., 2013*), usually in bipartite form (*e.g.* UAGNNUAG) (*Dominguez et al., 2018*), as the consensus binding motif for *msi-1*. There is a conserved bipartite UAG motif in the intron downstream of the *sad-1* cassette exon (*Figure 6G–H*), and we hypothesize that *msi-1* binds there to mediate exon inclusion in inhibitory motor neurons.

### Phenotypic convergence of splicing regulation in diverse neuron types

Together the results from three different neuronal cell types (ALM neuron, excitatory motor neurons, and inhibitory motor neurons) constitute an example of 'phenotypic convergence,' in which phenotypic similarity between cells is generated by distinct molecular mechanisms. Substantial evidence of such phenotypic convergence exists for TFs controlling neuronal properties in worms and flies (*Gendrel et al., 2016*; *Konstantinides et al., 2018*; *Pereira et al., 2015*). Our results now extend this principle to RBPs and their control of alternative splicing, revealing phenotypic convergence in which similar splicing patterns (*i.e. sad-1* exon inclusion) are generated in various neurons by diverse RBPs acting in specific neuronal subtypes (*Figure 7*).

To further examine this principle, we tested whether ectopic expression of an RBP in a neuron type in which it is not normally expressed would be sufficient to alter *sad-1* splicing in that neuron. We expressed *mec-8* in excitatory motor neurons (where normally only *mbl-1* is expressed) and found that *mec-8* expression is sufficient to alter *sad-1* splicing patterns from partial inclusion to full inclusion specifically in the excitatory motor neurons (*Figure 6—figure supplement 2A*). Similarly, *mbl-1* expression in inhibitory motor neurons (where normally only *msi-1* is expressed) results in full exon inclusion [*Figure 6—figure supplement 2A*]).

Finally, we asked whether phenotypic convergence occurs simultaneously at multiple levels (TFs and RBPs) with regard to *sad-1* splicing. To do so we examined mutants for the TF *unc-3*, which controls the fate of excitatory motor neurons in the ventral nerve cord (*Kratsios et al., 2012*), analogous to ALM cell fate determination by *unc-86/mec-3/alr-1*. In *unc-3* mutants, *sad-1* exon inclusion is lost in excitatory motor neurons, similar to *mbl-1* RBP mutants (*Figure 6D*, *Figure 6—figure supplement 2B*). However, whereas *unc-86/mec-3/alr-1* mutants exhibit completely-penetrant loss of *sad-1* exon inclusion, *unc-3* mutants exhibit partially-penetrant defects, ranging from moderate to complete loss of *sad-1* inclusion in excitatory motor neurons. Similarly, loss of *unc-3* results in partially-penetrant defects in *mbl-1* expression (*Figure 6—figure supplement 2C*).

Together these results demonstrate that phenotypic convergence among different neuron types occurs simultaneously at multiple layers of gene regulation: different TFs (*e.g. mec-3* and *unc-3*) specify expression of different RBP complements (*e.g. mbl-1* and *mec-8*) which have a common function of mediating *sad-1* exon inclusion.

## Discussion

### Neuron-specific regulation of *sad-1* splicing

In this study we find that *sad-1* splicing undergoes precise regulation in numerous neuronal types. Although ALM and BDU neurons are sister cells, express many of the same genes, and share a number of cell-specific TFs, they have opposing patterns of *sad-1* splicing. This highlights the fact that post-transcriptional control can further diversify attributes of single cells on top of the more well-known role of transcriptional control.

Our results demonstrate that *sad-1* splicing is regulated according to a combinatorial RBP code, with different splicing outcomes depending on whether a cell expresses zero, one, or two neuron-specific RBPs (*Figure 7*).This suggests that the 'default' outcome of *sad-1* splicing is full skipping of the cassette exon, as observed in the BDU neuron which does not express any of the *sad-1-*

regulating RBPs. Only cells with at least one RBP mediating exon inclusion express *sad-1* included isoforms. Cells with multiple such RBPs (*e.g.* the ALM neuron) express only the included isoform.

In previous work we found that alternative splicing of the kinase *unc-16/* JIP3 in motor neurons is likewise controlled by a pair of RNA binding proteins (*Norris et al., 2014*). However, *unc-16* splicing and *sad-1* splicing in motor neurons are regulated by distinct pairs of RBPs. Whereas *sad-1* splicing in motor neurons is regulated by *mbl-1* and *msi-1* RBPs, *unc-16* is combinatorially regulated by *unc-75* and *exc-7* in motor neurons (*Norris et al., 2014*). This suggests that even within a single neuron type, different splicing events are regulated by different complements of RBPs.

## Coordinated splicing regulation across layers of gene expression

The importance of TFs controlling gene expression networks in single neurons is well established, and the importance of RBPs controlling post-transcriptional networks in single cells is gaining wider appreciation (*Norris and Calarco, 2012*; *Norris et al., 2014*; *Song et al., 2017*; *Wamsley et al., 2018*). How these two modes of regulation might interact remains understudied. Here we show that the two modes of regulation interact in a traditional linear type of pathway. A combination of cell-specific TFs establishes a transcriptional network in a single neuron type. This network includes a specific combination of neuronal RBPs, and the particular combination of RBPs in a given neuron then establish a unique post-transcriptional gene regulatory network in that neuron. Multiple layers of regulatory control can thus increase the diversity of single neuron transcriptomes and fine-tune the properties of individual neurons.

In the present study we have identified a linear pathway in which TFs influence the expression of RBPs, which then influence alternative splicing in single neurons. This adds to a substantial body of literature finding that RBPs can affect the function of specific TFs by modulating their alternative splicing (*Calarco et al., 2009*; *Han et al., 2013*; *Linares et al., 2015*; *Raj et al., 2011*). In the future it will be interesting to see whether additional regulatory logics exist between TFs and RBPs. Single-neuron TF combinations have been identified with a variety of feedback and feedforward mechanisms resulting in interesting regulatory properties (*Mangan and Alon, 2003*), and in principle TFs and RBPs could likewise interact in complex ways, leading to an even greater array of diversification strategies (*Han et al., 2017*). Together this study highlights the importance of considering neuron-specific 'combinatorial codes' not only from the perspective of TF combinations, but the specific complement of both TFs and RBPs shaping the transcriptome of a given neuron.

## Phenotypic convergence at the level of single-neuron splicing

A theme emerging from recent studies of single-neuron transcriptomes is 'phenotypic convergence,' in which multiple neurons share gene expression similarities, but the regulatory mechanisms by which they do so are distinct in each neuron. For example, in worms, cholinergic neuron cell fate and core cholinergic gene expression properties are controlled by different combinations of TFs in different cholinergic neuron sub-types (*Pereira et al., 2015*). This is also the case for other neuron types in *C. elegans* (*Gendrel et al., 2016*). More recently, phenotypic convergence has been reported for TFs in neurons of the *Drosophila* optic lobe (*Konstantinides et al., 2018*), indicating that phenotypic convergence mediated by TFs is a widespread phenomenon.

We now extend this principle of phenotypic convergence to the regulation of splicing by RBPs as well. *sad-1* exon inclusion is mediated in various neuron types, with a unique complement of RBPs responsible for exon inclusion in each specific type that we have studied (ALM neuron, inhibitory motor neurons, and excitatory motor neurons; *Figure 7*). This likely represents phenotypic convergence on multiple levels, as the RBPs regulating splicing are different in each neuron, and the TFs regulating RBP expression are likewise different in each neuron. Each of these levels coordinately converges upon appropriate splicing of *sad-1* in each neuron type. Additional neuron types with similar *sad-1* splicing patterns (see *Figure 1* and *Figure 1—figure supplement 1*) may represent additional examples of phenotypic convergence whose underlying mechanisms remain unexplored.

# Materials and methods

## Key resources table

| Reagent type (species) or resource | Designation | Source or reference | Identifiers | Additional information |
|---|---|---|---|---|
| Strain | unc-86(csb9) | This study | JAC401 | Norris Lab. SMU. Dallas, TX. |
| Strain | mec-3(csb10) | This study | JAC402 | Norris Lab. SMU. Dallas, TX. |
| Strain | alr-1(csb11) | This study | JAC403 | Norris Lab. SMU. Dallas, TX. |
| Strain | unc-86(e1416) | CGC, University of Minnesota | CB1416 | |
| Strain | mec-3(e1338) | CGC, University of Minnesota | CB1338 | |
| Strain | alr-1(oy42) | CGC, University of Minnesota | PY1598 | |
| Strain | mec-8(e398) | CGC, University of Minnesota | CB398 | |
| Strain | mec-8(csb22) | This studdy | JAC626 | Norris Lab. SMU. Dallas, TX. |
| Strain | mbl-1(csb31) | This study | JAC635 | Norris Lab. SMU. Dallas, TX. |
| Strain | mbl-1(wy560) | CGC, University of Minnesota | JAC002 | |
| Strain | msi-1(csb24) | This study | JAC628 | Norris Lab. SMU. Dallas, TX. |
| Strain | mec-8(csb22); mbl-1(wy560) | This study | ADN342 | Norris Lab. SMU. Dallas, TX. |
| Strain | mec-8(csb22); mbl-1(csb31) | This study | JAC670 | Norris Lab. SMU. Dallas, TX. |
| Strain | msi-1(csb24); mbl-1(csb31) | This study | ADN257 | Norris Lab. SMU. Dallas, TX. |
| Strain | pmec-3::mec-8 | This study | ADN431 | Norris Lab. SMU. Dallas, TX. |
| Strain | pmec-3::mbl-1 | This study | ADN514 | Norris Lab. SMU. Dallas, TX. |
| Strain | punc-25::mbl-1 | This study | ADN515 | Norris Lab. SMU. Dallas, TX. |
| Strain | punc-17::mec-8 | This study | ADN505 | Norris Lab. SMU. Dallas, TX. |
| Strain | Δmbl-1 cis-element sad-1 splicing reporter | This study | ADN319 | Norris Lab. SMU. Dallas, TX. |
| Strain | Δmec-8[1] cis-element sad-1 splicing reporter | This study | ADN364 | Norris Lab. SMU. Dallas, TX. |
| Strain | Δmec-8[2] cis-element sad-1 splicing reporter | This study | ADN377 | Norris Lab. SMU. Dallas, TX. |
| Strain | Δmec-8[both] cis-element sad-1 splicing reporter | This study | ADN333 | Norris Lab. SMU. Dallas, TX. |
| Strain | sad-1 splicing reporter | This study | JAC017 | Norris Lab. SMU. Dallas, TX. |
| Strain | MEC-8::GFP reporter fosmid | This study | JAC447 | Norris Lab. SMU. Dallas, TX. |

*Continued on next page*

DOI: https://doi.org/10.7554/eLife.46726

*Continued*

| Reagent type (species) or resource | Designation | Source or reference | Identifiers | Additional information |
|---|---|---|---|---|
| Strain | MBL-1::RFP reporter fosmid | This study | JAC576 | Norris Lab. SMU. Dallas, TX. |
| Strain | *pmec-8::GFP* | CGC, University of Minnesota | BC11068 | |
| Strain | unc-3(e151) | CGC, University of Minnesota | CB151 | |

## Strain maintenance

*C. elegans* were maintained under standard conditions (*Brenner, 1974*) at 20°C on nematode growth media (NGM) plates seeded with OP50 *E. coli* bacteria. New transgenic worms were generated by microinjection with 15 ng/µl transgene and 15 ng/µl co-injection marker (either *rgef-1*, *unc-17*, or *unc-25* promoter driving BFP).

## Mutant generation and genetic screening

The forward mutagenesis screen was performed on animals harboring the *sad-1* exon 15 splicing reporter with EMS at 47 mM for 4 hr. $F_1$s were picked onto new plates, 10 $F_1$s per plate. After 3–4 days of growth, $F_2$s were screened by eye on the Zeiss Axiozoom.V16 for touch cells appearing in the GFP channel (representing aberrant exon skipping) and were then verified for a concomitant loss of RFP (representing loss of exon inclusion). Such worms were picked individually onto a new plate to verify the phenotype in the $F_3$ generation and to establish a clonal population. After outcrossing, strains were subjected to whole-genome resequencing (Illumina, 1 × 75 bp) and potential causative mutations were identified using the CloudMAP workflow on the Galaxy web platform (*Minevich et al., 2012*). A total of approximately 6000 haploid genomes were screened.

Targeted mutant strains were generated using CRISPR/Cas9 as previously described (*Calarco and Norris, 2018*; *Norris et al., 2015*), such that the gene of interest is deleted and is replaced with a heterologous GFP reporter under the control of a pharyngeal promoter (pmyo-2) which does not interfere with the visualization of the *sad-1* splicing reporter in the ALM, BDU or ventral nerve cord neurons. Seamless gene replacement was verified by PCR amplification and Sanger sequencing of both junction boundaries.

## Microscopy

Images were obtained with a Zeiss Axio Imager.Z1 and processed in ImageJ.

## Generation of splicing reporters

*sad-1* minigenes were created using the following primers: Forward 5' GATAAAACTGAAACAAC TTCTGC and Reverse 5' GGGGTTGGCGATTTGTATGAGaTAGC. Restriction sites were appended to both the forward primer (XhoI) and reverse (NotI) primers to facilitate cloning into a Gateway-compatible vector as previously described (*Norris et al., 2014*). The reporter was then cloned downstream of a pan-neuronal *rgef-1* promoter, as endogenous *sad-1* has been detected broadly throughout the nervous system (*Crump et al., 2001*). Mutant versions of the splicing reporter were synthesized de novo then cut with XhoI and NotI and cloned as above.

Some strains were provided by the Caenorhabditis Genome Center, which is funded by the NIH Office of Research Infrastructure Programs (P40 OD010440). Other strains were provided by the National BioResource Project (Tokyo).

## Acknowledgements

Some strains were provided by the CGC, which is funded by NIH Office of Research Infrastructure Programs (P40 OD010440). JAC is supported by funding from the Natural Sciences and Engineering Research Council of Canada (NSERC) and the Canadian Institutes of Health Research (CIHR), ADN is supported by funding from Oak Ridge Associated Universities (ORAU) and NIH R35 GM133461. Other strains were provided by the National BioResource Project (Tokyo). We thank Megan Norris

for critical reeding of the manuscript. MT devised experiments, performed experiments, wrote manuscript. RB performed experiments. RD performed experiments. AV performed experiments. JC devised experiments. AN devised experiments, performed experiments, wrote manuscript.

## Additional information

### Funding

| Funder | Grant reference number | Author |
|---|---|---|
| Canadian Institutes of Health Research | | John Calarco |
| Natural Sciences and Engineering Research Council of Canada | | John Calarco |
| Oak Ridge Associated Universities | | Adam D Norris |
| NIGMS | R35 GM133461 | Adam D Norris |

The funders had no role in study design, data collection and interpretation, or the decision to submit the work for publication.

### Author contributions

Morgan Thompson, Adam D Norris, Conceptualization, Investigation, Writing—original draft, Writing—review and editing; Ryan Bixby, Alexa Vandenburg, Investigation; Robert Dalton, Investigation, Writing—review and editing; John A Calarco, Conceptualization, Writing—review and editing

### Author ORCIDs

John A Calarco ⓘ https://orcid.org/0000-0002-2197-7801
Adam D Norris ⓘ https://orcid.org/0000-0002-0570-7414

### Decision letter and Author response

Decision letter https://doi.org/10.7554/eLife.46726.022
Author response https://doi.org/10.7554/eLife.46726.023

## Additional files

### Supplementary files

• Transparent reporting form
DOI: https://doi.org/10.7554/eLife.46726.020

### Data availability

All data from this study included in manuscript and supplemental materials.

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
