## [Decision Letter]

Thank you for submitting your article "Splicing in a single neuron is coordinately controlled by RNA binding proteins and transcription factors" for consideration by *eLife*. Your article has been reviewed by three peer reviewers, and the evaluation has been overseen by a Reviewing Editor and James Manley as the Senior Editor. The following individuals involved in review of your submission have agreed to reveal their identity: Hidehito Kuroyanagi (Reviewer #2).

The reviewers have discussed the reviews with one another and the Reviewing Editor has drafted this decision to help you prepare a revised submission.

Summary:

In this paper, the authors use an in vivo fluorescent splicing reporter to study cell-specific post-translational regulation. Making use of the genetic amenability of *C. elegans*, the authors employ forward genetic screens and candidate approaches to identify the genes required for cell-specific splicing events. Through the characterization of the complex splicing patterns of the conserved neuronal kinase *sad-1*, the authors elegantly reveal that neuronal identity regulator transcription factors and RNA binding proteins are required for *sad-1* exon inclusion. Neuron-type specific expression of RBPs is controlled by these transcription factors, and RBPs directly interact with *sad-1* introns. Moreover, the authors demonstrate that *sad-1* splicing in different neuron-types is controlled by different RBPs, revealing phenotypic convergence at the level of single-neuron splicing.

The authors employed CRISPR-Cas9 mediated genome editing and a smart splicing reporter to support their conclusions. The overall conclusions on transcriptional control of RBPs and phenotypic convergence appear interesting. However, there is some lack in the depth of analysis, particularly as it pertains to the analysis of the transcription factors. Moreover, there is striking, in fact almost baffling, lack of detail in data/reagent presentation throughout the manuscript. A substantial revision of the paper will allow the authors to better support what appears to be a very interesting finding.

Essential revisions:

1) There is a concern that the evidence that the three RBPs "directly" regulate pre-mRNA splicing of the *sad-1* gene via their respective cis-elements is weak. The conclusion is based on high-throughput in vitro screening for consensus binding sequences of the RBPs (Ray et al., 2013), effects of the RBP mutations on the *sad-1* reporter expression (Figure 3 and Figure 6) and effects of mutations in the putative cis-elements on the reporter expression (Figure 5). Direct and specific binding of recombinant RBPs to the respective RNA sequences in vitro and/or defective responses of the mutant reporters to RBP overexpression need to be provided support the direct regulation; otherwise the possibility that MBL-1 regulating *sad-1* splicing through post-transcriptional regulation of MEC-8 and/or other RBP(s), and vice versa, cannot be excluded. This point is important especially for *mbl-1* and *msi-1* because the manuscript will be the first experimental demonstration that these genes regulate pre-mRNA splicing in *C. elegans*. It is also important to keep in mind that the consensus sequences experimentally identified in Ray et al., 2013 and logos shown in Figure 3, Figure 5 and Figure 6 of the manuscript are for a human orthologue of MBL-1, human and *Drosophila* orthologues of MSI-1 and not those in *C. elegans* (this should be more explicitly stated in the figures).

2) The common standard for validating that a mutant retrieved from a screen really is the causative mutant is to either rescue or use multiple alleles. The authors do neither. They need to validate the identity of their screen mutants by examining the expression of the *sad-1* splicing reporter on a canonical allele of at least *mec-3*.

3) It would be appreciated if the authors could extend their concept of TFs controlling splicing patterns by RBPs through testing whether *mbl-1* expression in excitatory neurons – and hence *sad-1* splicing – depends on *unc-3*, the excitatory MN "alter ego" of *unc-86/mec-3* in the touch neurons. This experiment is very easy to perform and would broaden the concept of cell type-specific TFs controlling cell-type specific RBPs to other neuron types.

4) Figure 7. The authors discuss that 100% exon inclusion might require multiple RBPs. Alternatively, 100% exon inclusion might require RBP binding on both sides of the exon in question (e.g. in ALM, *mbl-1* seems to bind upstream of *sad-1* exon 15, while *mec-8* seems to bind downstream). This hypothesis can be easily tested by *mec-8* overexpression in cholinergic MN (mirroring ALM neurons), or by *mbl-1* overexpression in GABAergic MN (preferable since it is a new untested scenario).

5) Generally, there is a baffling, if not troubling lack of any detail in describing and properly quantifying data and of description of reagents. This pervades the entire manuscript. Below a number of examples (in no order of importance):

a) In panels Figure 2C-F, Figure 3D-G, Figure 4B-E, Figure 4F-I, Figure 5, Figure 6C-F, Figure 3—figure supplement 2, Figure 4—figure supplement 1, Figure 4—figure supplement 2, and Figure 4—figure supplement 3 the authors do not provide any quantification to support their data (incl. the number of animals examined). It is unclear whether in all cases there is a fully penetrant phenotype. In cases of a fully penetrant phenotype (e.g. Figure 2C-F?), a simple explanation on the legend accompanied with the number of animals where the phenotype was observed will suffice. It is odd that the authors do not appear to be familiar with the standards in the field. In the case of partial exon skipping phenotypes, it is unclear to this reviewer whether all animals show always same levels of GFP and RFP expression in ALM (e.g. in the double heterozygotes). For variable phenotypes, a qualitative quantification (%dark/dim/bright red/green), or any other means of quantification deemed appropriate by the authors, is required.

b) There's no description of the splicing reporter. Is the reporter driven by its own promoter or the endogenous promoter? They refer to a previous paper (Norris, 2014) which states vectors that drove expression of each reporter pan-neuronally, or with endogenous promoter sequences

c) The authors generate a number of deletion mutants for different genes using CRISPR/Cas9 (*mec-8, mbl-1*, not specified for *msi-1*) however, they don't provide any description on the nature of the molecular lesion other than a brief reference to a previous paper. A schematic of the mutations should be provided in the respective main figures or in a supplementary figure, with a more detailed explanation on the legends or in the Materials and methods section.

d) There is no list of the strains used on this study as a supplementary file. For example, which is the allele name of the new deletions?

e) Authors need to describe how new transgenic strains were generated (DNA mix concentrations, simple or complex arrays, etc.).

f) These newly generated deletion mutants via CRISPR/Cas9 seem to include a GFP reporter construct, subsection “Mutant generation and genetic screening”: "the gene of interest is deleted and is replaced with a heterologous GFP reporter construct". However, authors use fosmid reporters to analyze *mec-8* and *mbl-1* expression (Figure 4). Do these new *mbl-1* and *mec-8* deletion alleles recapitulate *mbl-1* and *mec-8* expression through the GFP reporter? In that case, does this GFP expression interfere with the analysis of the *sad-1* splicing reporter green fluorescence?

g) Subsection “RBPs directly mediate *sad-1* exon inclusion through interactions with surrounding introns”, "Most neurons besides the ALM and BDU neurons express both skipped and included *sad-1* isoforms". Attending to the merge provided in Figure 1C, several neurons in the head and tail seem to exclude exon 15. It will be very informative if authors provide a more detailed expression data of their splicing reporter on Figure 1. For example, just a gross quantification of the% of neurons with included/excluded/both in the head, tail, major ganglia, etc., will greatly help the authors to better illustrate the splicing event under study.

h) Figure 4A-E, *mec-8* and *mbl-1* expression. Was *mec-8* fosmid reporter expression detected in other neurons? Also, authors mention that the *mbl-1* fosmid reporter is expressed in "most other neurons" (subsection “TFs affecting *sad-1* splicing are required for RBP expression in the ALM neuron”, Figure 4D). To support their point, authors could show whole worm images or high magnification images of specific regions (head, tail, etc.). This needs to be properly documented with an image.

---

## [Author Response]

Essential revisions:

*1) There is a concern that the evidence that the three RBPs "directly" regulate pre-mRNA splicing of the sad-1 gene via their respective cis-elements is weak. The conclusion is based on high-throughput* in vitro *screening for consensus binding sequences of the RBPs (Ray et al., 2013), effects of the RBP mutations on the sad-1 reporter expression (Figure 3 and Figure 6) and effects of mutations in the putative cis-elements on the reporter expression (Figure 5). Direct and specific binding of recombinant RBPs to the respective RNA sequences* in vitro *and/or defective responses of the mutant reporters to RBP overexpression need to be provided support the direct regulation; otherwise the possibility that MBL-1 regulating sad-1 splicing through post-transcriptional regulation of MEC-8 and/or other RBP(s), and vice versa, can not be excluded. This point is important especially for mbl-1 and msi-1 because the manuscript will be the first experimental demonstration that these genes regulate pre-mRNA splicing in C. elegans. It is also important to keep in mind that the consensus sequences experimentally identified in Ray et al., 2013 and logos shown in Figure 3, Figure 5 and Figure 6 of the manuscript are for a human orthologue of MBL-1, human and Drosophila orthologues of MSI-1 and not those in C. elegans (this should be more explicitly stated in the figures).*

We have performed the suggested in vivoexperiments, subjecting mutant reporters to RBP overexpression, and found that the defective splicing pattern remains unchanged (subsection “RBPs directly mediate *sad-1* exon inclusion through interactions with surrounding introns” and Figure 5—figure supplement 1). We also now explicitly state in the figures which species provided each recombinant RBP for generation of in vitrobinding motifs.

2) The common standard for validating that a mutant retrieved from a screen really is the causative mutant is to either rescue or use multiple alleles. The authors do neither. They need to validate the identity of their screen mutants by examining the expression of the sad-1 splicing reporter on a canonical allele of at least mec-3.

We thank the reviewers for pointing out this oversight. We have remedied the error, displaying canonical alleles for each of the TFs identified in our screen in Figure 2 (with allele designations in legend), and alleles identified from our screen in Figure 2—figure supplement 1.

3) It would be appreciated if the authors could extend their concept of TFs controlling splicing patterns by RBPs through testing whether mbl-1 expression in excitatory neurons – and hence sad-1 splicing – depends on unc-3, the excitatory MN "alter ego" of unc-86/mec-3 in the touch neurons. This experiment is very easy to perform and would broaden the concept of cell type-specific TFs controlling cell-type specific RBPs to other neuron types.

These experiments have been performed, and support the general concept of TFs controlling alternative splicing via RBP expression (Figure 6—figure supplement 2). One unique aspect we have noted is that with *unc-3* mutants and excitatory motor neurons the phenotypes seem to be less penetrant than the fully-penetrant defects in *unc-86/mec-3/alr-1* mutants (discussed in the Discussion section).

4) Figure 7. The authors discuss that 100% exon inclusion might require multiple RBPs. Alternatively, 100% exon inclusion might require RBP binding on both sides of the exon in question (e.g. in ALM, mbl-1 seems to bind upstream of sad-1 exon 15, while mec-8 seems to bind downstream). This hypothesis can be easily tested by mec-8 overexpression in cholinergic MN (mirroring ALM neurons), or by mbl-1 overexpression in GABAergic MN (preferable since it is a new untested scenario).

Both experiments have been performed (Figure 6—figure supplement 2, and subsection “Phenotypic convergence of splicing regulation in diverse neuron types”), and indeed, ectopic expression of either *mbl-1* or *mec-8* in a neuron type in which they are not normally expressed is sufficient to alter *sad-1* splicing patterns (in both cases changing the neuron’s splicing pattern from partial exon inclusion to ALM-like full inclusion).

5) Generally, there is a baffling, if not troubling lack of any detail in describing and properly quantifying data and of description of reagents. This pervades the entire manuscript. Below a number of examples (in no order of importance):a) In panels Figure 2C-F, Figure 3D-G, Figure 4B-E, Figure 4F-I, Figure 5, Figure 6C-F, Figure 3—figure supplement 2, Figure 4—figure supplement 1, Figure 4—figure supplement 2, and Figure 4—figure supplement 3 the authors do not provide any quantification to support their data (incl. the number of animals examined). It is unclear whether in all cases there is a fully penetrant phenotype. In cases of a fully penetrant phenotype (e.g. Figure 2C-F?), a simple explanation on the legend accompanied with the number of animals where the phenotype was observed will suffice. It is odd that the authors do not appear to be familiar with the standards in the field. In the case of partial exon skipping phenotypes, it is unclear to this reviewer whether all animals show always same levels of GFP and RFP expression in ALM (e.g. in the double heterozygotes). For variable phenotypes, a qualitative quantification (%dark/dim/bright red/green), or any other means of quantification deemed appropriate by the authors, is required.

We apologize for the lack of various details, including quantification, in the first draft of the manuscript. Quantification has now been added in figure legends for all figures requested, as well as a few additional figures. This quantification demonstrates that most (but not all) conditions result in fully-penetrant and invariant splicing outcomes.

b) There's no description of the splicing reporter. Is the reporter driven by its own promoter or the endogenous promoter? They refer to a previous paper (Norris, 2014) which states vectors that drove expression of each reporter pan-neuronally, or with endogenous promoter sequences

This has been added in subsection “Microscopy”.

c) The authors generate a number of deletion mutants for different genes using CRISPR/Cas9 (mec-8, mbl-1, not specified for msi-1) however, they don't provide any description on the nature of the molecular lesion other than a brief reference to a previous paper. A schematic of the mutations should be provided in the respective main figures or in a supplementary figure, with a more detailed explanation on the legends or in the Materials and methods section.

This has been added as Figure 3—figure supplement 1

d) There is no list of the strains used on this study as a supplementary file. For example, which is the allele name of the new deletions?

This has been added in the Key Resource table at the beginning of the Materials and methods section.

e) Authors need to describe how new transgenic strains were generated (DNA mix concentrations, simple or complex arrays, etc.).

We have now included details of our microinjection conditions with DNA concentrations, subsection “Strain maintenance”. For all experiments, we generated simple arrays.

f) These newly generated deletion mutants via CRISPR/Cas9 seem to include a GFP reporter construct, subsection “Mutant generation and genetic screening”: "the gene of interest is deleted and is replaced with a heterologous GFP reporter construct". However, authors use fosmid reporters to analyze mec-8 and mbl-1 expression (Figure 4). Do these new mbl-1 and mec-8 deletion alleles recapitulate mbl-1 and mec-8 expression through the GFP reporter? In that case, does this GFP expression interfere with the analysis of the sad-1 splicing reporter green fluorescence?

This has been clarified (subsection “Mutant generation and genetic screening”) to explain that the GFP in the CRISPR alleles does not report on the endogenous expression of the targeted gene, but rather uses a heterologous *myo-2* pharyngeal promoter to express GFP, in effect “marking” the deletion allele. Given that the GFP is expressed only in the pharynx, this does not interfere with analysis of the *sad-1* reporter in the ALM of VNC neurons.

g) Subsection “RBPs directly mediate sad-1 exon inclusion through interactions with surrounding introns”, "Most neurons besides the ALM and BDU neurons express both skipped and included sad-1 isoforms". Attending to the merge provided in Figure 1C, several neurons in the head and tail seem to exclude exon 15. It will be very informative if authors provide a more detailed expression data of their splicing reporter on Figure 1. For example, just a gross quantification of the% of neurons with included/excluded/both in the head, tail, major ganglia, etc., will greatly help the authors to better illustrate the splicing event under study.

This has been added in Figure 1—figure supplement 1.

h) Figure 4A-E, mec-8 and mbl-1 expression. Was mec-8 fosmid reporter expression detected in other neurons?

We have added text (subsection “TFs affecting *sad-1* splicing are required for RBP expression in the ALM neuron”) that *mec-8* expression was observed in other neurons in addition to ALM.

Also, authors mention that the mbl-1 fosmid reporter is expressed in "most other neurons" (subsection “TFs affecting sad-1 splicing are required for RBP expression in the ALM neuron”, Figure 4D). To support their point, authors could show whole worm images or high magnification images of specific regions (head, tail, etc.). This needs to be properly documented with an image.

We have added a whole worm image for reference (Figure 4—figure supplement 1).